# Model-free tracking control of complex dynamical trajectories with machine learning

Zheng-Meng Zhai [1], Mohammadamin Moradi [1], Ling-Wei Kong [1], Bryan Glaz[2], Mulugeta Haile [3] & Ying-Cheng Lai [1,4] ✉

Nonlinear tracking control enabling a dynamical system to track a desired trajectory is fundamental to robotics, serving a wide range of civil and defense applications. In control engineering, designing tracking control requires complete knowledge of the system model and equations. We develop a model-free, machine-learning framework to control a two-arm robotic manipulator using only partially observed states, where the controller is realized by reservoir computing. Stochastic input is exploited for training, which consists of the observed partial state vector as the first and its immediate future as the second component so that the neural machine regards the latter as the future state of the former. In the testing (deployment) phase, the immediate-future component is replaced by the desired observational vector from the reference trajectory. We demonstrate the effectiveness of the control framework using a variety of periodic and chaotic signals, and establish its robustness against measurement noise, disturbances, and uncertainties.

The traditional field of controlling chaotic dynamical systems mostly deals with the problem of utilizing small perturbations to transform a chaotic trajectory into a desired periodic one[1]. The basic principle is that the dynamically invariant set that generates chaotic motions contains an infinite number of unstable periodic orbits. For any desired system performance, it is often possible to find an unstable periodic orbit whose motion would produce the required behavior. The problem then becomes one to stabilize the system's state-space or phase-space trajectory around the desired unstable periodic orbit, which can be achieved through linear control in the vicinity of the orbit, thereby requiring only small control perturbations. The control actions can be calculated from the locations and the eigenvalues of the target orbit, which are often experimentally accessible through a measured time series, without the need to know the actual system equations[1–4]. Controlling chaos can thus be done in a model-free, entirely data-driven manner, and the control is most effective when the chaotic behavior is generated by a low-dimensional invariant set, e.g., one with one unstable dimension or one positive Lyapunov exponent. However, for high-dimensional dynamical systems, controlling complex nonlinear dynamical networks is an active area of research[5–7].

The goal of tracking control is to design a control law to enable the output of a dynamical system (or a process) to track a given reference signal. For linear feedback systems, tracking control can be mathematically designed with rigorous guarantee of stability[8]. However, nonlinear tracking control is more challenging, especially when the goal is to make a system to track a complex signal. In robotics, for instance, a problem is to design control actions to make the tip of a robotic arm, or the end effector, to follow a complicated or chaotic trajectory. In control engineering, designing tracking control typically requires complete knowledge of the system model and equations. Existing methods for this include feedback linearization[9], backstepping control[10], Lyapunov redesign[11], and sliding mode control[12]. These classic nonlinear control methods may face significant challenges when dealing with high-dimensional states, strong nonlinearity

[1]School of Electrical, Computer and Energy Engineering, Arizona State University, Tempe, AZ 85287, USA. [2]Army Research Directorate, DEVCOM Army Research Laboratory, 2800 Powder Mill Road, Adelphi, MD 20783-1138, USA. [3]Army Research Directorate, DEVCOM Army Research Laboratory, 6340 Rodman Road, Aberdeen Proving Ground, MD 21005-5069, USA. [4]Department of Physics, Arizona State University, Tempe, AZ 85287, USA. ✉e-mail: Ying-Cheng.Lai@asu.edu

or time delays[13,14], especially when the system model is inaccurate or unavailable. Developing model-free and purely data-driven nonlinear control methods is thus at the forefront of research. In principle, data-driven control has the advantage that the controller is able to adjust in real-time to new dynamics under uncertain conditions, but existing controllers are often not sufficiently fast "learners" to accommodate quick changes in the system dynamics or control objectives[15]. In this regard, tracking a complex or chaotic trajectory requires that the controller be a "fast responder" as the target state can change rapidly. At the present, developing model-free and fully data-driven control for fast tracking of arbitrary trajectories, whether simple or complex (ordered or chaotic), remains to be an challenging problem. This paper aims to address this challenge by leveraging recent advances in machine learning.

Recent years have witnessed a rapid expansion of machine learning with transformative impacts across science and engineering. This progress has been fueled by the availability of vast quantities of data in many fields as well as by the commercial success in technology and marketing[15]. In general, machine learning is designed to generate models of a system from data. Machine-learning control is of particular relevance to our work, where a machine-learning algorithm is applied to control a complex system and generate an effective control law that maps the desired system output to the input. More specifically, for complex control problems where an accurate model of the system is not available, machine learning can leverage the experience and data to generate an effective controller. Earlier works on machine-learning control concentrated on discrete-time systems, but the past few years have seen growing efforts in incorporating machine learning into control theory for continuous-time systems in various applications[16–19].

There are four types of problems associated with machine-learning control: control parameter identification, regression based control design of the first kind, regression based control design of the second kind, and reinforcement learning. For control parameter identification, the structure of the control law is given but the para-meters are unknown, an example of which is developing genetic algorithms for optimizing the coefficients of a classical controller [e.g., PID (proportional-integral-derivative) control or discrete-time optimal control[20,21]]. For regression-based control design of the first kind, the task is to use machine learning to generate an approximate nonlinear mapping from sensor signals to actuation commands, an example of which is neural-network enabled computation of sensor feedback from a known full state feedback[22]. For regression-based control design of the second kind, machine learning is exploited to identify arbitrary nonlinear control laws that minimize the cost function of the system. In this case, it is not necessary to know the model, control law structure, or the optimizing actuation command, and optimization is solely based on the measured control performance (cost function), for which genetic programming represents an effective regression technique[23,24]. For reinforcement learning, the control law can be continually updated over measured performance changes based on rewards[25–32]. It should be noted that historically, reinforcement learning control is not always model free. For instance, an early work[33] proposed a model-based learning method for nonlinear control where the basic idea is to decompose a complex task into multiple domains in space and time based on the predictability of the dynamics of the environment. A framework was developed[34,35] to determine both the feedback and feed-forward components of the control input simultaneously, enabling reinforcement learning to solve the tracking problem without requiring complete knowledge of the system dynamics and leading to the on- and off-policy algorithms[36].

Since our aim is to achieve tracking control of complex and chaotic trajectories, a natural choice of the machine-learning frame-work is reservoir computing[37–39] that has been demonstrated to be powerful for model-free prediction of nonlinear and chaotic systems[40–53]. The core of reservoir computing is recurrent neural network (RNN) with low training cost where regularized linear regression is sufficient for training. Reservoir computing, shortly after its invention, was exploited to control dynamical systems[54] where an inverse model was trained to map the present state and the desired state of the system to the control signal (action). Subsequently, the trained reservoir computer was exploited as a model-free nonlinear feedback controller[55] as well as for detecting unstable periodic orbits and stabilizing the system about a desired orbit[56]. Reservoir computing and its variant echo state Gaussian process[57] were also used in model predictive control of unknown nonlinear dynamical systems[58,59], which served as replacements of the traditional recurrent neural-network models with low computational cost. More recently, deep reservoir networks were proposed for controlling chaotic systems[60].

In this paper, we tackle the challenge of model-free and data-driven nonlinear tracking of various reference trajectories, including complex chaotic trajectories, with an emphasis on their potential applications in robotics. In particular, we examine the case of a two-arm robotic manipulator with the control objective of tracking any trajectories while using only partially observed states, denoted as vector $\mathbf{y}(t)$. Our control framework has the following three features: (1) requirement of only partial state observation for both training and testing, (2) a machine-learning training scheme that involves the observed vectors at two consecutive time steps: $\mathbf{y}(t)$ and $\mathbf{y}(t + dt)$, and (3) use of a stochastic signal as the input control signal for training. With respect to feature (1), it may be speculated that the classical Takens delay-coordinate embedding methodology could be used to construct the full phase space from partial observation. However, in this case, the reconstructed state is equivalent to the original system but only in a topological sense: there is no exact state correspondence between the reconstructed and the original dynamical systems. For reservoir-computing based prediction and control tasks, such an exact correspondence is required. To our knowledge, achieving tracking control based on partial state observation is novel. In terms of features (2) and (3), we note a previous work[55] on machine-learning stabilization of linear and low-dimensional nonlinear dynamical systems, where the phase-space region to realize control is localized. This was effectively an online learning approach. In general, online learning algorithms have difficulties such as instability, modeling complexity as required for nonlinear control, and computational efficiency. For example, it is difficult for online learning to capture the intricate complex nonlinear dynamics, causing instability during control. Trajectory divergence is another common problem associated with online learning control, where sudden and extreme changes in the state can occur. In fact, as the dimension and complexity of the system to be controlled increase, online learning algorithms tend to fail. In contrast, offline learning is computationally extremely efficient and allows for more comprehen-sive and complex model training with minimum risk of trajectory divergence through repeated training. Our tracking framework entails following a dynamic and time-varying (even chaotic) trajectory in the whole phase space, where the offline controller can not only respond to disturbances and system variations but also adjust the control inputs to make the system output follow a continuously changing reference signal. As we will demonstrate, our control scheme brings these features together to enable continuous tracking of arbitrary complex trajectories.

## Results

A more detailed explanation of the three features and their combina-tion to solve the complex trajectory tracking problem is as follows. First, existing works on reservoir-computing based controllers relied on full state measurements[54–56,58–60], but our controller requires mea-suring only a partial set of the state variables. Second, as shown in Fig. 1a, during the training phase, the input to the machine learning controller consists of two components: the observation vector at two consecutive time steps: $\mathbf{y}(t)$ and $\mathbf{y}(t + dt)$. That is, at any time step $t$, the

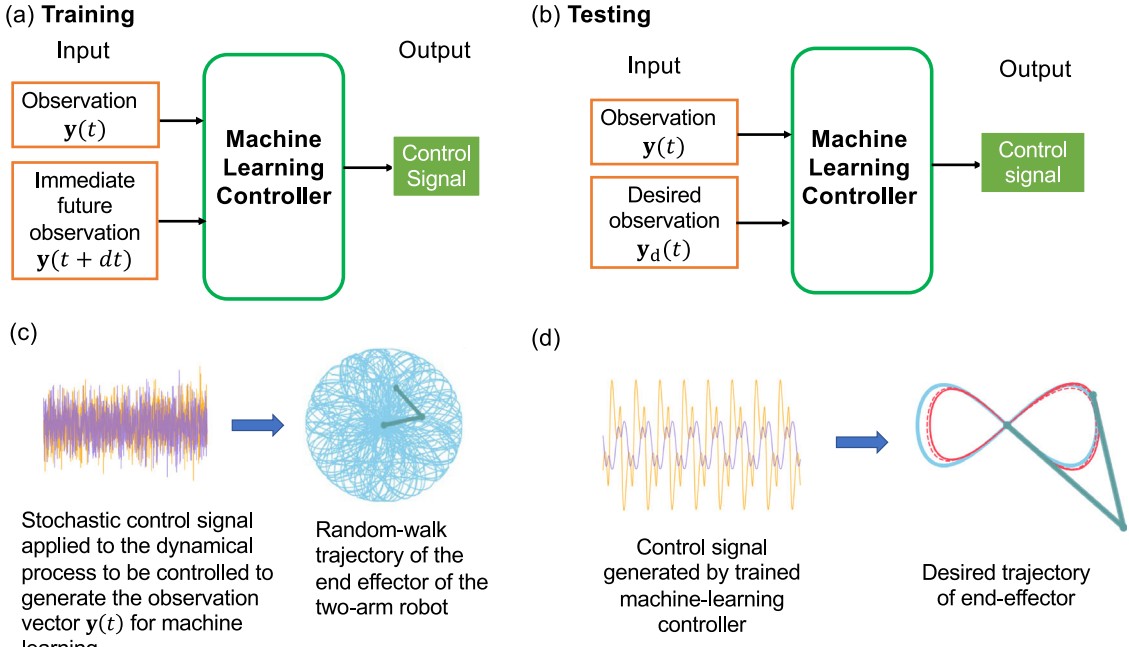

**Fig. 1 | Illustration of our proposed machine-learning tracking controller.**
**a** During the training phase, the input to the machine-learning controller consists of two vectors of equal dimension: the partial observation vector $\mathbf{y}(t)$ and its immediate-future counterpart $\mathbf{y}(t+dt)$ as the complementary. The output is a control signal which, when applied to the dynamical system or process, will enable it to track any desired reference trajectory. This input configuration stipulates that the complementary component of the input is the immediate future state of the observation vector. **b** In the testing phase, the complementary component of the input vector is replaced by $\mathbf{y}_d(t)$, the observation vector calculated from the reference trajectory. Since the machine-learning controller has been trained to recognize the complementary input component as the immediate future state, in the testing phase the controller will "force" the observation vector to follow the desired vector, thereby realizing accurate tracking. Note that $\mathbf{y}_d(t)$ is provided to the machine-learning controller according to the desired trajectory, so no process model is required. **c** A fully stochastic control signal is used for training, which generates a random-walk type of reference trajectory. The required input vectors $\mathbf{y}(t)$ and $\mathbf{y}(t+dt)$ to machine learning are obtained by observing the dynamical process to be controlled, so a mathematical model of the process is not required. **d** A well-trained machine learning controller generates the appropriate control signal to track any desired trajectory, where the blue and dotted red traces correspond to the reference and tracked trajectories, respectively.

second vector is the state of the observation vector in the immediate future. This input configuration offers several advantages, which are evident in the testing phase, as shown in Fig. 1b. After the machine-learning controller has been trained, the testing input consists of the observation vector $\mathbf{y}(t)$ and the desired observation vector $\mathbf{y}_d(t)$, calculated from the reference trajectory to be tracked. The idea is that, during the testing or deployment, the immediate future state of the observation is manipulated to match the desired vector from the trajectory. This way, the output control signal from the machine-learning controller will make the end effector of the robotic manipulator to precisely trace out the desired reference trajectory. The third feature is the choice of the control signal for training. Taking advantage of the fundamental randomness underlying any chaotic trajectory, we conduct the training via a completely stochastic control input, as shown in Fig. 1c, where the reference trajectory generated by such a control signal through the underlying dynamical process is a random walk. Compared with a deterministic chaotic trajectory with short-term predictability, the random-walk trajectory is more complex as its movements are completely unpredictable. As a result, the machine-learning controller trained with a stochastic signal will possess a level of complexity sufficient for controlling or overpowering any deterministic chaotic trajectory. In general, our machine-learning controller so trained is able to learn a mapping between the state error and a suitable control signal for any reference trajectory. In the testing phase, given the current and desired states, the machine-learning controller generates the control signal that enables the robotic manipulator to track any desired complex reference trajectory, as illustrated in Fig. 1d. We demonstrate the working and power of our machine-learning tracking control using a variety of periodic and chaotic trajectories, and establish the robustness against measurement noise, disturbances, and uncertainties. While our primary machine-learning scheme is reservoir computing, we also test the architecture of feed-forward neural networks and demonstrate its working as an effective tracking controller, albeit with higher computational time complexity. Overall, our work provides a powerful model-free data-driven control framework that only relies on partial state observation and can successfully track complex or chaotic trajectories.

## Principle of machine-learning based control

An overview of the working principle of our machine-learning based tracking control is as follows. Consider a dynamical process to be controlled, e.g., a two-arm robotic system, as indicated in the green box on the left in Fig. 2. The objective of control is to make the end effector, which is located at the tip of the outer arm, track a complex trajectory. Let $\mathbf{x} \in \mathcal{R}^D$ represent the full, $D$-dimensional state space of the process. An observer has access to part of the full state space and produces a $D'$-dimensional measurement vector $\mathbf{y}$, where $D' < D$. A properly selected and trained machine-learning scheme takes $\mathbf{y}$ as its input and generates a low-dimensional control signal $\mathbf{u}(t) \in \mathcal{R}^{D''}$ (e.g., two respective torques applied to the two arms), where $D'' \le D'$, to achieve the control objective. The workings of our control scheme can be understood in terms of the following three essential components: (1) a mathematical description of the dynamical process and the observables (Methods), (2) a physical description of how to obtain the control signals from the observables (known as inverse dynamics—Methods) and (3) the machine-learning scheme (Supplementary Note 1).

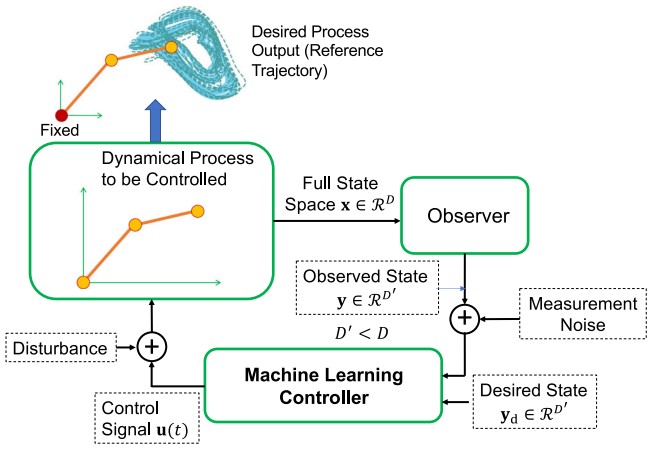

**Fig. 2 | Working principle of our machine-learning based tracking control.** The state space vector **x** of the dynamical process to be controlled is $D$-dimensional. An observer produces a $D'$-dimensional measurement vector **y**, where $D'<D$. The machine-learning controller uses this vector and the corresponding desired vector **y**$_d$ calculated from the reference trajectory to be tracked as the input and generates a proper, typically lower-dimensional control signal **u**$(t)$. Disturbance is applied to the control signal vector **u** and measurement noise is present during the observation of the state vector **x**. Unlike controllers that rely on the error between **y** and **y**$_d$, our controller uses both signals as inputs, which provides it with two degrees of freedom.

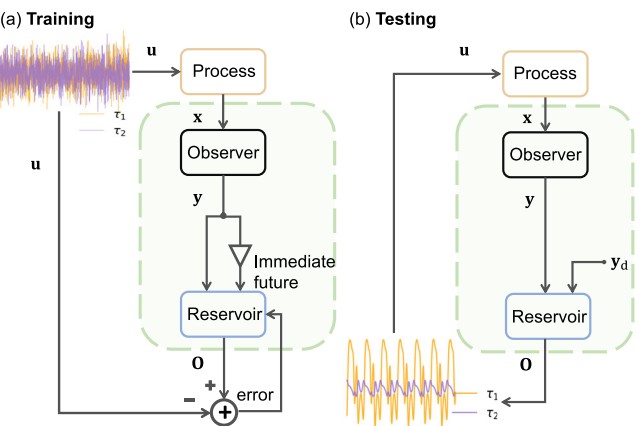

**Fig. 3 | Basic architecture of proposed machine-learning based tracking control framework for the two-arm robotic system. a** During the training phase, the random torques $\tau_1(t)$ and $\tau_2(t)$ are used as the control signal to the dynamical process (the two-arm system) to generate the state vector **x**$(t)$. An observer produces a lower-dimensional observed vector **y**$(t)$. This vector and its immediate future **y**$(t+dt)$ are used as the input to the reservoir computing machine, whose output is a two-dimensional torque vector. The difference between the reservoir output and the original random torque signal constitutes the error for training. **b** In the testing or deployment phase, the input to the reservoir computer is the observed vector **y**$(t)$ and the desired vector **y**$_d(t)$ calculated from the reference trajectory. The output of the reservoir is a control signal that drives the two-arm system so that its end effector precisely traces out the desired reference trajectory.

The state variable of the two-joint robot-arm system is eight-dimensional: $\mathbf{x} \equiv [C_x, C_y, q_1, q_2, \dot{q}_1, \dot{q}_2, \ddot{q}_1, \ddot{q}_2]^T$, where $C_x$ and $C_y$ are the Cartesian coordinates of the end effector, $q_i$, $\dot{q}_i$ and $\ddot{q}_i$ are the angular position, angular velocity and angular acceleration of aim $i$ ($i=1, 2$). The measurement vector is four-dimensional: $\mathbf{y} \equiv [C_x, C_y, \dot{q}_1, \dot{q}_2]^T$. A remarkable feature of our framework is that a purely stochastic signal can be conveniently used for training. As illustrated in Fig. 1c, the torques $\tau_1(t)$ and $\tau_2(t)$ applied to the two arms, respectively, are taken to be stochastic signals from a uniform distribution, which produce a random-walk type of trajectory of the end effector. The control input for training is $\mathbf{u}(t)=[\tau_1(t),\tau_2(t)]^T$, as shown in Fig. 3a. To ensure continuous control input, we use a Gaussian filter to smooth the noise input data. With the control signal, the forward model Eq. (13) (in Methods) produces the state vector **x**$(t)$ and the observer generates the vector **y**$(t)$. The observed vector **y**$(t)$ and its delayed version **y**$(t+dt)$ constitute the input to the reservoir computing machine that generates a control signal **O**$(t)$ as the output, leading to the error signal $\mathbf{e}(t) = \mathbf{O}(t) - \mathbf{u}(t)$ as the loss function for training the neural network.

A well trained reservoir can then be tested or deployed to generate any desired control signal, as illustrated in Fig. 3(b). In particular, during the testing phase, the input to the reservoir computer consists of the observed vector **y**$(t)$ and the desired vector **y**$_d(t)$ characterized by the two Cartesian coordinates of the reference trajectory of the end effector and the resulting angular velocities of the two arms. Note that, given an arbitrary reference trajectory $\{C_x(t), C_y(t)\}$, the two angular velocities can be calculated (extrapolated) from Eqs. (8) and (9) (in Methods). The output of the reservoir computing machine is the two required torques $\tau_1(t)$ and $\tau_2(t)$ that drive the two-arm system so that the end effector traces out the desired reference trajectory.

**Training.** The detailed structure of the data and the dynamical variables associated with the training process is described, as follows. The training phase is divided into a number of uncorrelated episodes, each of length $T_{ep}$, which defines the resetting time. At the start of each episode, the state variables including $[\dot{q}_1, \dot{q}_2, \ddot{q}_1, \ddot{q}_2]$ along with the controller state are reset. The initial angular positions $q_1$ and $q_2$ are randomly chosen in their defined range, respectively. For each episode, the process's control input is stochastic for a time duration of $T_{ep}$,

generating a torque matrix of dimension $2 \times T_{ep}$, as illustrated in Fig. 4. For the same time duration, the state **x** of the dynamical process and the observed state **y** can be expressed as a $8 \times T_{ep}$ and a $4 \times T_{ep}$ matrix, respectively. At each time step $t$, the input to the reservoir computing machine, the concatenation of **y**$(t)$ and **y**$(t+dt)$, is an $8 \times 1$ vector. The neural network learns to generate a control input that takes the process's output from **y**$(t)$ to **y**$(t+dt)$ so as to satisfy the tracking goal. The resulting trajectory of the end effector of the process, due to the stochastic input torques, is essentially a random walk. To ensure that the random walk covers as much of the state space as possible, the training length and machine-learning parameters must be appropriately chosen.

**Testing.** In the testing phase, the trained neural network inverts the dynamics of the process. In particular, given the current and desired output, the neural network generates the control signal to drive the system's output from **y**$(t)$ to **y**$(t+dt)$ while minimizing the error between **y**$(t+dt)$ and **y**$_d(t+dt)$. We shall demonstrate that our machine-learning controller is capable of tracking any complicated trajectories, especially a variety of chaotic trajectories.

With a reservoir controller and the inverse model, our tracking-control framework is able to learn the mapping between the current and desired position of the end effector and deliver a proper control signal, for a given reference trajectory. For demonstration, we use 16 different types of reference trajectories including those from low- and high-dimensional chaotic systems. (The details of the generation of these reference trajectories are presented in Supplementary Note 2) Note that the starting position of the end effector is not on the given reference trajectory, requiring a "bridge" to drive the end effector from the starting position to the trajectory (See Supplementary Note 3). Here we also address the issue of probability of control success and the robustness of our method against measurement noise, disturbance, and parameter uncertainties.

## Examples of tracking control
The basic parameter setting of the reservoir controller is as follows. The size of the hidden-layer network is $N_r = 200$. The dimensionless

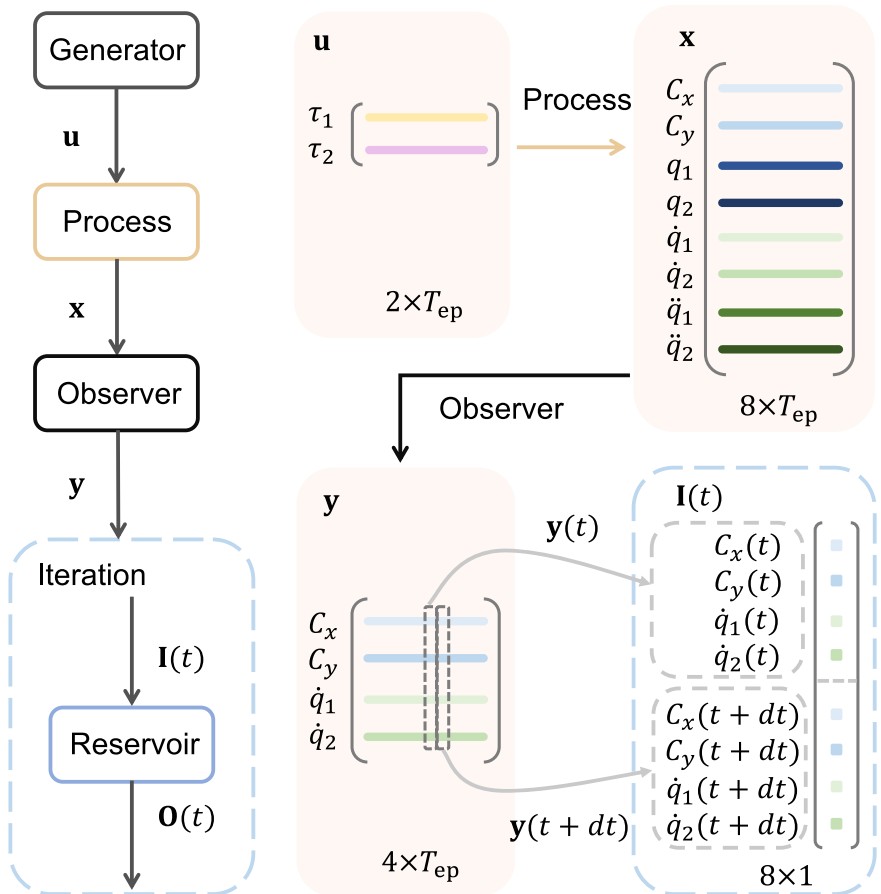

**Fig. 4 | Dynamical variables and data structure associated with the training phase of the machine-learning controller.** Two stochastic signals act as torques for the two-arm system, causing its end effector to generate a random walk. For a training episode of duration $T_{ep}$, the input to the process is a $2 \times T_{ep}$ data matrix. The state and the observed vectors are represented as a $8 \times T_{ep}$ and a $4 \times T_{ep}$ matrix, respectively. At any time step $t$, the input to the reservoir computing machine is an eight-dimensional vector constituting $\mathbf{y}(t)$ and $\mathbf{y}(t+dt)$.

time step of the evolution of dynamical network is $dt = 0.01$. A long training length is chosen: $200,000/dt$ so as to ensure that the learning experience of the neural network extends through most of the phase space in which the reference trajectory resides. The testing length is $2,500/dt$, which is sufficient for the controller to track a good number of complete cycles of the reference trajectory. The values of the reservoir hyperparameters obtained through Bayesian optimization are: spectral radius $\rho = 0.76$, input weights factor $\gamma = 0.76$, leakage parameter $\alpha = 0.84$, regularization coefficient $\beta = 7.5 \times 10^{-4}$, link probability $p = 0.53$, and the bias $w_b = 2.00$.

The training phase is divided into a series of uncorrelated episodes, ensuring that the velocity or acceleration of the robot arms will not become unreasonably large during the random-walk motion of the reference trajectory. The episodes are initialized at time $T_{ep} = 80/dt$. The angular positions $q_1$ and $q_2$ of the two arms is set to a random value uniformly distributed in the ranges $[0, 2\pi]$ and $[-\pi, \pi]$, respectively. The angular velocities and accelerations $[\dot{q}_1, \dot{q}_2, \ddot{q}_1, \ddot{q}_2]$ of the two arms as well as the reservoir state $\mathbf{r}$ are set to zero initially. From the values of $q_1$ and $q_2$, the coordinates $C_x$ and $C_y$ of the end effector can be obtained from Eq. (7). At the beginning of each episode, since $q_1$ and $q_2$ are random, the end-effector will be a random point inside a circle of radius $l_1 + l_2 = 1$ centered at the origin. Figure 5a shows the random-walk reference trajectory used in training and examples of the evolution of the dynamical states of the two arms (in two different colors): $q_{1,2}(t), \dot{q}_{1,2}(t), \ddot{q}_{1,2}(t)$, and $\tau_{1,2}(t)$. To maintain the continuity of the control signal during the training phase, we invoke a Gaussian filter to smooth the noisy signals. Given the control signal $\mathbf{u}(t) = [\tau_1(t), \tau_2(t)]$ and

the state variables $[q_{1,2}(t), \dot{q}_{1,2}(t)]$ at each time step, the angular accelerations $\ddot{q}_{1,2}(t)$ can be obtained from Eq. (4). At the next time step, the angular positions and velocities are calculated using

$$q_{1,2}(t+dt) = q_{1,2}(t) + \dot{q}_{1,2}(t) \cdot dt,$$
$$\dot{q}_{1,2}(t+dt) = \dot{q}_{1,2}(t) + \ddot{q}_{1,2}(t) \cdot dt. \tag{1}$$

The purpose of the training is for the reservoir controller to learn the intrinsic mapping from $\mathbf{y}(t)$ to $\mathbf{y}(t+dt)$ and to produce an output control signal $\mathbf{u}(t) = [\tau_1(t), \tau_2(t)]$.

In the testing phase, given the current measurement $\mathbf{y}(t)$ and the desired measurement $\mathbf{y}_d(t+dt)$, the reservoir controller generates a control signal and feed it to the process. The tracking error is the difference between $\mathbf{y}_d(t+dt)$ and $\mathbf{y}(t+dt)$. Figure 5(b) presents four examples: two chaotic (Lorenz and Mackey-Glass) and two periodic (a circle and an eight figure) reference trajectories, where in each case, the angular positions, velocities, and accelerations of both arms together with the control signal (the two torques) delivered by the reservoir controller are shown. As the reservoir controller has been trained to track a random walk signal, which is fairly complex and chaotic, it possesses the ability to accurately track these types of deterministic signals.

Our machine-learning controller, by design, is generalizable to arbitrarily complex trajectories. This can be seen, as follows. In the training phase, no specific trajectory is used. Rather, training is accomplished by using a stochastic control signal to generate a random-walk type of trajectory that "travels" through the entire state-

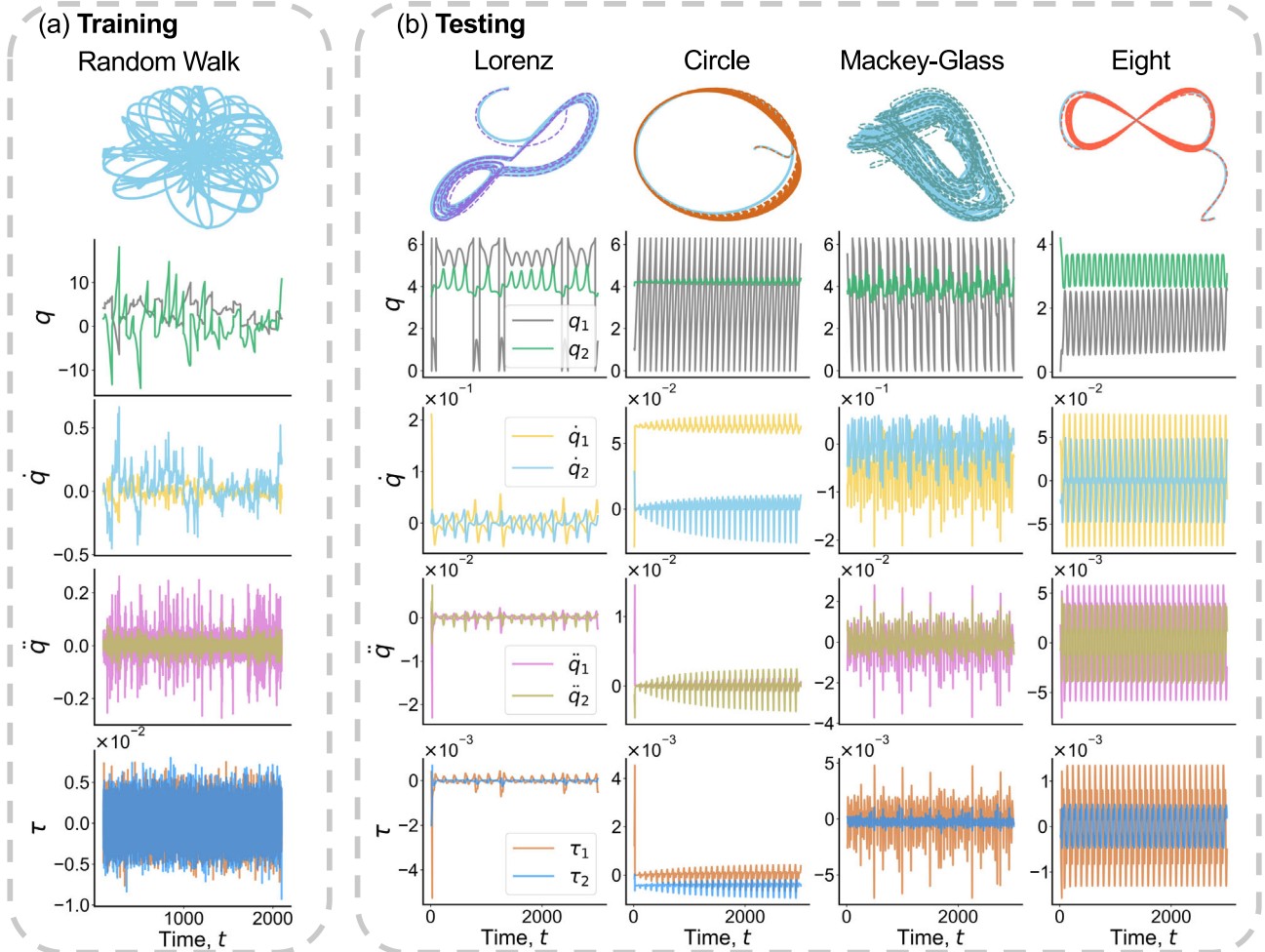

**Fig. 5 | Examples of tracking control. a** A random-walk reference trajectory used in training. The time series plots below are the angular positions ($q$), velocities ($\dot{q}$), and accelerations ($\ddot{q}$) as well as the two torques (the control signals, $\tau$) applied to the two arms. **b** Successful tracking of four reference trajectories: two chaotic (Lorenz and Mackey-Glass) and two periodic (a circle and an eight figure). Solid blue and dotted traces represent the reference and controlled trajectories, respectively. The reservoir controller generates the proper control signals based on the current measurement vector $\mathbf{y}(t)$ and the corresponding desired vector $\mathbf{y}_d(t)$.

space domain of interest. The machine-learning controller does not learn any specific trajectory example but a generic map from the observed state at the current time step to the next under a stochastic control signal. The training process determines the parameter values for the controller, which are fixed when it is deployed in the testing phase. The required input for testing is the current observed state $\mathbf{y}(t)$ and the desired state $\mathbf{y}_d(t)$ from the reference trajectory. The so-designed machine-learning controller is capable of making the system to follow a variety of complex periodic or chaotic trajectories to which the controller is not exposed during training. (Supplementary Notes 2 and 4 present many additional examples).

**Robustness against disturbance and noise**

We consider normally distributed stochastic processes of zero mean and standard deviations $\sigma_d$ and $\sigma_m$ to simulate disturbance and noise, which are applied to the control signal vector $\mathbf{u}$ and the process state vector $\mathbf{x}$, respectively, as shown in Fig. 2. Fig. 6(a) and (b) show the ensemble-averaged testing RMSE (root mean square error, defined in Supplementary Note 1) versus $\sigma_d$ and $\sigma_m$, respectively, for tracking of the chaotic Lorenz reference trajectory, where 50 independent realizations are used to calculate the average errors. In the case of disturbance, near zero RMSEs are achieved for $\sigma_d \lesssim 10^{0.5}$, while the noise tolerance is about $10^{-1}$. Color-coded testing RMSEs in the parameter plane ($\sigma_d$, $\sigma_m$) are shown in Fig. 6(c). Those results indicate that, for

reasonably weak disturbances and small noise, the tracking performance is robust. (Additional examples are presented in Supplementary Note 4).

**Robustness against parameter uncertainties**

The reservoir controller is trained for ideal parameters of the dynamical process model. However, in applications, the parameters may differ from their ideal values. For example, the lengths of the two robot arms may deviate from what the algorithm has been trained for. More specifically, we narrow our attention to the uncertainty associated with the arm lengths, as variations in the mass parameters do not noticeably impact the control performance. Figure 7 shows the results from the uncertainty test in tracking a chaotic Lorenz reference trajectory. It can be seen that changes in the length $l_1$ of the primary arm have little effect on the performance. Only when the length $l_2$ of the secondary arm becomes much larger than $l_1$ will the performance begin to deteriorate. The results suggest that our control framework is able to maintain good performance if the process model parameters are within reasonable limits. In fact, when the lengths of the two robot arms are not equal, there are reference trajectories that the end-effector cannot physically track. For example, consider a circular trajectory of radius $l_1 + l_2$. For $l_2 < l_1$, it is not possible for the end effector to reach the points in the circle of radius $l_1 - l_2$. More results from the parameter-uncertainty test can be found

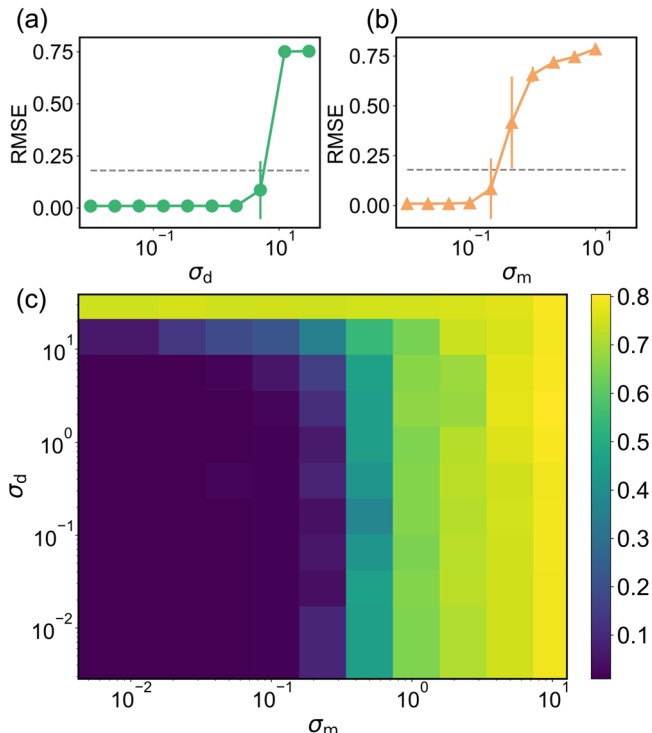

**Fig. 6 | Robustness against disturbance and noise for tracking the chaotic Lorenz reference trajectory. a**, **b** Ensemble-averaged testing RMSE versus the amplitude $\sigma_d$ of the disturbance and the noise amplitude $\sigma_m$, respectively. Error bars represent standard deviation calculated from 50 independent realizations. In each case, the horizontal dashed line represents some empirical threshold below which the tracking-control performance may be regarded as satisfactory. The tolerance of tracking control to disturbance is about $\sigma_d \lesssim 10^{0.5}$ and that to noise is about $\sigma_m \lesssim 10^{-1}$. **c** Color-coded RMSE in the parameter plane ($\sigma_d$, $\sigma_m$).

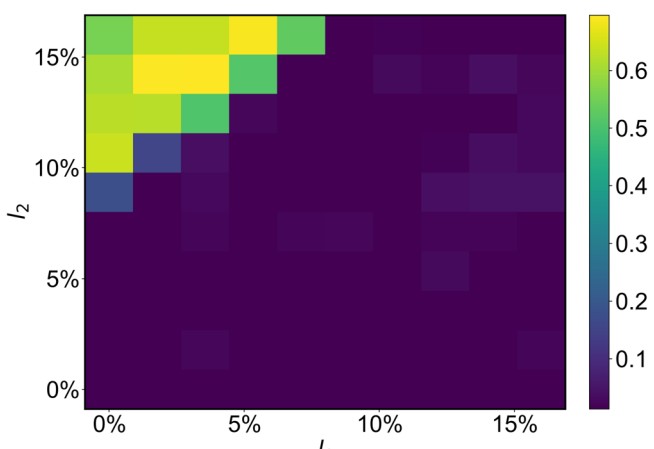

**Fig. 7 | Robustness against parameter uncertainties in the process model.** Shown is a color map of the testing RMSE in the parameter plane of the lengths ($l_1$, $l_2$) of the two arms, where the chaotic Lorenz trajectory in Fig. 5b is used as the reference. The ideal model parameters used in the training are $l_1 = 0.5$ and $l_2 = 0.5$. Each RMSE value is the result of averaging over 50 independent realizations. The RMSE values are small for most of the parameter region, and the performance of the reservoir controller is especially robust against the uncertainty in the length of the primary arm.

in Supplementary Note 4. The issues of safe region of initial conditions for control success, tracking speed tolerance, and robustness against variations in training parameters are addressed in Supplementary Note 5.

## Discussion

The two main issues in control are: (1) regularization, which involves designing a controller so that the corresponding closed-loop system converges to a steady state, and (2) tracking - to make the output of the closed-loop system track a given reference trajectory continuously. In both cases, the goal is to achieve optimal performance despite disturbances and initial states[61]. The conventional method for control systems design is linear quadratic tracker (LQT), whose objective is, e.g., to design an optimal tracking controller by minimizing a pre-defined performance index. Solutions to LQT in general consist of two components: a feedback term obtained by solving an algebraic Riccati equation and a feed-forward term which is obtained by solving a non-causal difference equation. These solutions require complete knowledge of the system dynamics and cannot be obtained in real time[62]. Another disadvantage of LQT is that it can be used only for the class of reference trajectories generated by an asymptotically stable command generator that requires the trajectory to approach zero asymptotically. Furthermore, the LQT solutions are typically non-causal due to the necessity of backward recursion, and the infinite horizon LQT problem is challenging in control theory[63]. The rapidly growing field of robotics requires the development of real-time, non-LQT solutions for tracking control.

We have developed a real-time nonlinear tracking control method based on machine learning and partial state measurements. The benchmark system employed to illustrate the methodology is a two-arm robotic manipulator. The goal is to apply appropriate control signals to make the end effector of the manipulator to track any complex trajectory in a 2D plane. We have exploited reservoir computing as the machine-learning controller. With proper training, the reservoir controller acquires inherent knowledge about the dynamical system generating the reference trajectory. Our inverse controller design method requires the observed state vector and its immediate future as input to the neural network in the training phase. The testing or deployment phase requires a combination of the current and desired output measurements: no future measurements are needed. More specifically, in the training phase, the input to the reservoir neural network consists of two vectors of equal dimension: (a) the observed vector from the robotic manipulator and (b) its immediate future version. This design enables the controller to naturally associate the second vector with the immediate future state of the first vector in the testing phase and to generate control signals based on this association. After training, the parameters of the machine-learning controller are fixed for testing, which distinguishes our control scheme from online learning. The controller in the testing phase is deployed to track a desired reference trajectory since the immediate future vectors $\mathbf{y}(t + dt)$ are replaced by the states generated from the desired reference trajectory, which are recognized by the machine as the desired immediate future states of the robotic manipulator to be controlled. The control signal generated in this manner compels the manipulator to imitate the dynamical system that generates the reference trajectory, resulting in precise tracking. We also take advantage of stochastic control signals for training the neural network to enable it to gain as much dynamical complexity as possible.

We have tested this reservoir computing based tracking control using a variety of periodic and chaotic reference trajectories. In all the cases, accurate tracking for an arbitrarily long period of time can be achieved. We have also demonstrated the robustness of our control framework against input disturbance, measurement noise, process parameter uncertainties, and variations in the machine-learning parameters. A finding is that selecting the starting end-effector position "wisely" can improve the tracking success rate. In particular, we have introduced the concept of "safe region" from which the initial position of the end effector should be chosen (Supplementary Note 5). In addition, the effects of the amplitude of the stochastic control signal used in training and of the "speed limit" of the reference trajectory on

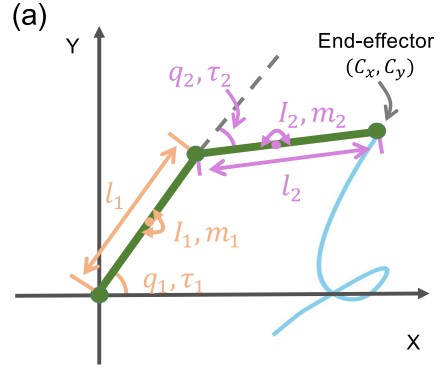
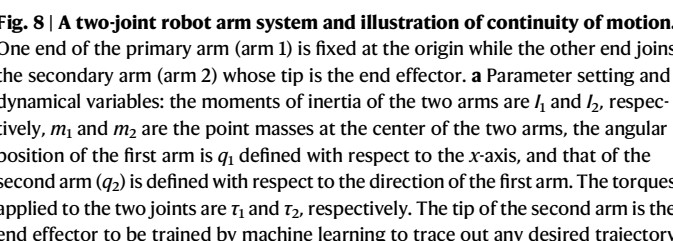
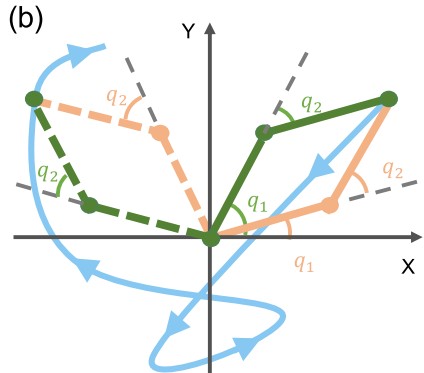

**Fig. 8 | A two-joint robot arm system and illustration of continuity of motion.** One end of the primary arm (arm 1) is fixed at the origin while the other end joins the secondary arm (arm 2) whose tip is the end effector. **a** Parameter setting and dynamical variables: the moments of inertia of the two arms are $I_1$ and $I_2$, respectively, $m_1$ and $m_2$ are the point masses at the center of the two arms, the angular position of the first arm is $q_1$ defined with respect to the x-axis, and that of the second arm ($q_2$) is defined with respect to the direction of the first arm. The torques applied to the two joints are $\tau_1$ and $\tau_2$, respectively. The tip of the second arm is the end effector to be trained by machine learning to trace out any desired trajectory (the blue curve). **b** Illustration of continuity of motion of the two arms: two possible configurations of the arms (green and orange, respectively) and a trajectory (blue). For the orange configuration, initially the angle $q_2$ is positive because the second arm is above the line extending the first arm. After going through the motion as specified by the blue trajectory, the final angular position of the second arm is still positive. For the green configuration, the initial angle $q_2$ is negative and it remains to be negative after the motion. It is necessary to calculate the angles from Eqs. (8) and (9) to satisfy the continuity condition.

the tracking success rate have been investigated (Supplementary Note 5). We have also demonstrated that feed-forward neural networks can be used to replace reservoir computing (Supplementary Note 6). The results suggest the practical utilities of our machine-learning based tracking controller: it is anticipated to be deployable in real-world applications such as unmanned aerial vehicle, soft robotics, laser cutting, soft robotics, and real-time tracking of high-speed air launched effects.

Finally, we remark that there are traditional methods for tracking control, such as PID, MPC (model predictive control), and $H\infty$ trackers (see refs. 20,21, references therein). In terms of computational complexity, these classical controllers are extremely efficient, while the training of our machine-learning controller with stochastic signals can be quite demanding. However, there is a fundamental limitation with the classic controllers: such a controller can be effective only when its parameters were meticulously tuned for a specific reference trajectory. For a different trajectory, a completely different set of parameters is needed. That is, when the parameters of a classic controller are set, in general it cannot be used to track any alternative trajectory. In contrast, our machine-learning controller overcomes this limitation: it possesses the remarkable capability and flexibility to track any given trajectory after a single training session! This distinctive attribute sets our approach apart from conventional methods, so a direct comparison with these methods may not be meaningful.

## Methods
### Dynamics of joint robot arms
The dynamics of the system of n-joint robot arms can be conveniently described by the standard Euler-Lagrangian method[64]. Let $T$ and $U$ be the kinetic and potential energies of the system, respectively. The equations of motion can be determined from the system Lagrangian $L = T - U$ as

$$\frac{d}{dt}\frac{\partial L}{\partial \dot{\mathbf{q}}} - \frac{\partial L}{\partial \mathbf{q}} = \boldsymbol{\tau}, \tag{2}$$

where $\mathbf{q} = [q_1, q_2, \ldots q_n]^T$ and $\dot{\mathbf{q}} = [\dot{q}_1, \dot{q}_2, \ldots, \dot{q}_n]^T$ are the angular position and angular velocity vectors of the n arms [with $()^T$ denoting the transpose], and $\boldsymbol{\tau} = [\tau_1, \tau_2, \ldots, \tau_n]^T$ is the external force vector with each

component applied to a distinct joint denoted by the subscript n. The nonlinear dynamical equations for the robot-arm system can be expressed as[65,66]

$$\mathcal{M}(\mathbf{q})\ddot{\mathbf{q}} + C(\mathbf{q},\dot{\mathbf{q}})\dot{\mathbf{q}} + \mathbf{G}(\mathbf{q}) + \mathbf{F}(\dot{\mathbf{q}}) = \boldsymbol{\tau}, \tag{3}$$

where $\ddot{\mathbf{q}} = [\ddot{q}_1, \ddot{q}_2, \ldots, \ddot{q}_n]^T$ is the acceleration vector of the n joints, $M(\mathbf{q})$ denotes the inertial matrix, $C(\mathbf{q},\dot{\mathbf{q}})\dot{\mathbf{q}}$ represents the Coriolis and centrifugal force, $\mathbf{G}(\mathbf{q})$ is the gravitational force vector, and $\mathbf{F}(\dot{\mathbf{q}})$ is the vector of the frictional forces at the n joints which depends on the angular velocities. We assume that the movements of the robot arms are confined to the horizontal plane so that the gravitational forces can be disregarded, and we also neglect the frictional forces, so Eq. (3) becomes

$$\mathcal{M}(\mathbf{q})\ddot{\mathbf{q}} + C(\mathbf{q},\dot{\mathbf{q}})\dot{\mathbf{q}} = \boldsymbol{\tau}. \tag{4}$$

We focus on the system of two joint robot arms ($n = 2$), as shown in Fig. 8, where $m_1$ and $m_2$ are the centers of the mass of the two arms, $l_1$ and $l_2$ are their lengths, respectively. The tip of the second arm is the end effector to trace out a desired trajectory in the plane. The two matrices in Eq. (4) are

$$\mathcal{M}(\mathbf{q}) = \begin{bmatrix} M_{11} & M_{12} \\ M_{21} & M_{22} \end{bmatrix} \tag{5}$$

$$\mathcal{C}(\mathbf{q},\dot{\mathbf{q}}) = \begin{bmatrix} -h(\mathbf{q})\dot{q}_2 & -h(\mathbf{q})(\dot{q}_1 + \dot{q}_2) \\ h(\mathbf{q})\dot{q}_1 & 0 \end{bmatrix}, \tag{6}$$

where the matrix elements are given by

$$M_{11} = m_1 l_{c_1}^2 + I_1 + m_2(l_1^2 + l_{c_2}^2 + 2l_1 l_{c_2} \cos q_2) + I_2,$$
$$M_{12} = M_{21} = m_2 l_1 l_{c_2} \cos q_2 + m_2 l_{c_2}^2 + I_2,$$
$$M_{22} = m_2 l_{c_2}^2 + I_2,$$

the function $h(\mathbf{q})$ is

$$h(\mathbf{q}) = m_2 l_1 l_{c_2} \sin q_2,$$

$l_{c_1} = l_1/2, l_{c_2} = l_2/2, I_1$ and $I_2$ are the moments of inertia of the two arms, respectively. Typical parameter values are $m_1 = m_2 = 1$, $l_1 = l_2 = 0.5, l_{c_1} = l_{c_2} = 0.25$, and $I_1 = I_2 = 0.03$.

The Cartesian coordinates of the end effector are

$$C_x = l_1 \cos q_1 + l_2 \cos(q_1 + q_2),$$
$$C_y = l_1 \sin q_1 + l_2 \sin(q_1 + q_2), \quad (7)$$

which give the angular positions of the two arms as

$$q_2 = \pm \arccos \frac{C_x^2 + C_y^2 - l_1^2 - l_2^2}{2l_1 l_2}, \quad (8)$$

$$q_1 = \arctan \frac{C_y}{C_x} \pm \arctan \frac{l_2 \sin q_2}{l_1 + l_2 \cos q_2}. \quad (9)$$

For any end-effector position, there are two admissible solutions for the angular variables. We select the pair of angles that result in a continuous trajectory. In addition, the end effector may end up in any of the four quadrants, so the range of $q_1$ is $[0, 2\pi]$. The range of $q_2$ is $[-\pi, \pi]$, since the second joint can be above or below the first joint. In our simulations, we ensure that the solutions are continuous and thus are physically meaningful, as demonstrated in Fig. 8b.

Noises and unpredictable disturbances are constantly present in real-world applications, making it crucial to ensure that the control strategy is robust and operational in their presence[67]. In fact, a model is always inaccurate compared with the actual physical system because of factors such as change of parameters, unknown time delays, measurement noise, and input disturbances. The goal of the robustness test is to maintain an acceptable level of performance under these circumstances. In our study, we treat disturbances and measurement noise as external inputs, where the former are added to the control signal and the latter is present in the sensor measurements. In particular, the disturbances are modeled as an additive stochastic process $\xi$ to the data:

$$\tilde{x}_n = x_n + \xi_d. \quad (10)$$

For measurement noise, we use multiplicative noise $\xi$ in the form

$$\tilde{x}_n = x_n + x_n \cdot \xi_m. \quad (11)$$

Both stochastic processes $\xi_d$ and $\xi_m$ follow a normal distribution of zero mean and with standard deviation $\sigma_d$ and $\sigma_m$, respectively.

**Inverse design based controller formulation**

To develop a machine-learning based control method, it is necessary to obtain the control signal through observable states. The state of the two-arm system, i.e., the dynamical process to be controlled, is eight-dimensional, which consists of the Cartesian coordinates of the end-effector, the angular positions, angular velocities and angular accelerations of the two manipulators:

$$\mathbf{x} \equiv [C_x, C_y, q_1, q_2, \dot{q}_1, \dot{q}_2, \ddot{q}_1, \ddot{q}_2]^T. \quad (12)$$

A general nonlinear control problem can be formulated as[60]

$$\mathbf{x}(t + dt) = \mathbf{f}[\mathbf{x}(t), \mathbf{u} + \mathbf{u} \cdot \xi_d], \quad (13)$$

$$\mathbf{y}(t) = \mathbf{g}[\mathbf{x}(t)] + \mathbf{g}[\mathbf{x}(t)] \cdot \xi_m, \quad (14)$$

where $\mathbf{x} \in \mathbb{R}^n$ $(n = 8)$, $\mathbf{u} \in \mathbb{R}^m$ $(m < n)$ is the control signal, $\mathbf{y} \in \mathbb{R}^k$ $(k \leq n)$ represents the sensor measurement. The function $\mathbf{f}$ :

$\mathbb{R}^n \times \mathbb{R}^m \to \mathbb{R}^n$ is unknown for the controller. In our analysis, we assume that $\mathbf{f}$ is Lipschitz continuous[68] with respect to $\mathbf{x}$. The measurement function $\mathbf{g} : \mathbb{R}^n \to \mathbb{R}^k$ fully or partially measures the states $\mathbf{x}$. For the two-arm system, the measurement vector is chosen to be four-dimensional: $\mathbf{y} \equiv [C_x, C_y, \dot{q}_1, \dot{q}_2]^T$. The corresponding vector from the desired, reference trajectory is denoted as $\mathbf{y}_d(t)$. For our tracking control problem, the aim is to design a two-degree-of-freedom controller that receives the signals $\mathbf{y}(t)$ and $\mathbf{y}_d(t)$ as the input and generates an appropriate control signal $\mathbf{u}(t)$ in order for $\mathbf{y}(t)$ to track the trajectory generating the observation $\mathbf{y}_d(t)$. For convenience, we use the notation $\mathbf{f}_u(\cdot) \equiv \mathbf{f}(\cdot, \mathbf{u})$. For a small time step $dt$, Eq. (13) becomes

$$\mathbf{x}(t + dt) \approx \mathbf{F}_u[\mathbf{x}(t)], \quad (15)$$

where $\mathbf{F}_u$ is a nonlinear function mapping $\mathbf{x}(t)$ to $\mathbf{x}(t + dt)$ under the control signal $\mathbf{u}(t)$. For reachable desired state, $\mathbf{F}_u$ is invertible. We get

$$\mathbf{u}(t) \approx \mathbf{F}_u^{-1}[\mathbf{x}(t), \mathbf{x}(t + dt)], \quad (16)$$

Similarly, Eq. (14) can be approximated as $\mathbf{x}(t) \approx \mathbf{g}^{-1}[\mathbf{y}(t)]$, so Eq. (16) becomes

$$\mathbf{u}(t) \approx \mathbf{F}^{-1}[\mathbf{g}^{-1}[\mathbf{y}(t)], \mathbf{g}^{-1}[\mathbf{y}(t + dt)]]. \quad (17)$$

Equation (17) is referred to as the inverse model for nonlinear control[60], which will be realized in a model-free manner using machine learning.

## Data availability

The reference trajectories data generated in this study can be found in the repository: https://doi.org/10.5281/zenodo.8044994[69].

## Code availability

The codes for generating all the results can be found on GitHub: https://github.com/Zheng-Meng/TrackingControl[70].

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

## Acknowledgements

This work was supported by the Army Research Office through Grant No.W911NF-21-2-0055 (to Y.-C.L.) and by the Air Force Office of Scientific Research through Grant No. FA9550-21-1-0438 (to Y.-C.L.).

## Author contributions

Z.-M.Z., M.M, L.-W.K., B.G., M.H. and Y.-C.L. designed the research project, the models, and methods. Z.-M.Z. performed the computations. Z.-M.Z., M.M., L.-W.K., B.G., M.H. and Y.-C.L. analyzed the data. Z.-M.Z. and Y.-C.L. wrote the paper. M.H. and Y.-C.L. edited the manuscript.

## Competing interests

The authors declare no competing interests.
