## [Peer Review File · Nature Communications]

REVIEWER COMMENTS

Reviewer #1 (Remarks to the Author):

The aim of this study is to develop a model-free nonlinear control method for tracking periodic and chaotic motions of a two-arm robot. Unlike traditional control methods relying on prior knowledge and equations about the target system to be controlled, the authors construct a machine learning model to predict appropriate control inputs from observed states of a target system. They demonstrate that the proposed method works well for tracking complex trajectories generated by nonlinear and chaotic dynamical systems. Simulation results show that the method is tolerant against reasonably weak noise and disturbance.

The basic idea behind the proposed model is found in the previous works [54,55] on tracking control, where an inverse model mapping a system state to a control input is approximated by a reservoir computing model. In this study, some advances are achieved. First, the controller works with data obtained by partial observations, by which the effort for measurements of system states can be reduced. Second, a pair of the present state and the one-step-ahead state is used as the input to the reservoir, which is new and reasonable for learning the association between them. Third, stochastic signals are used as control inputs in the training phase. The numerical results show that these factors are beneficial in successful tracking control.

The manuscript is well written, and the descriptions are informative. The methods and results are interesting overall, but there are some points to be addressed.

(1) The authors call " $y(t+dt)$ " time-delayed observation. I don't know why this is called "time-delayed" because a time-delayed state normally indicates " $y(t-dt)$ " in dynamical systems theory. In addition, it is questionable how $y(t+dt)$, which has not yet observed at time t , can be used for the update of the system at time t . It is better to explain how the observation process and training process are performed in practice, e.g. the order of the processes. If a storage of observed states is necessary, it might be a demerit of the proposed method compared to other online control methods. This point should be discussed.

(2) As shown in Fig. 1(c), the stochastic signals are used as control inputs. To learn appropriate associations between the present state and the one-step-ahead state, a large amount of training data (i.e. a long training length) seems to be necessary for the random-walk trajectory of the two-arm robot to fully cover the two-dimensional space. It is unclear how the authors determine the training length and how the training length affects the tracking performance. In addition, it is not obvious how much

generalization ability of the model is acquired by the proposed method, i.e. whether the trained model has the ability to track unknown trajectory that was not used in the training phase.

(3) The authors demonstrate that the proposed method works well for 15 different kinds of target dynamical systems in the manuscript and the supplementary file. All these target systems are low-dimensional nonlinear dynamical systems and therefore can be regarded as toy problems, although the experiments are exhaustive. It is better to show that the proposed method is applicable to more practical case. It is also desirable to compare the proposed method with other tracking control methods in terms of tracking accuracy and computational efficiency.

Reviewer #2 (Remarks to the Author):

"Model-free tracking control of complex dynamical trajectories with machine learning" studies the use of reservoir computing for control applications. A reservoir computer is used to predict a necessary control signal to steer a target dynamical system into a desired trajectory. Here, the authors use noise in training, and demonstrate the process in a large set of target dynamics.

The main idea is sound and clearly works, despite some insufficient explanations in the supplementary materials. The result can be useful for many applications, but I worry about the novelty of the result. The authors identify 3 major aspects:

- (1) use of only partial observations
- (2) time-delayed input configuration for machine learning
- (3) stochastic signals as the input control signals

While the combination of these aspects is new, individually these have already been explored. It is known that reservoir computers can infer the state of unobserved variables, such as in the case of partial observations. For example, the work cited as Ref. 40 in the present manuscript presents such a case.

On the other hand, the basic training setup as shown in Fig 1 seems identical to that presented in Figure 2 of "Feedback Control by Online Learning an Inverse Model

" (2012) by Waegeman, wyffels and Schrauwen, which is also cited as Reference 55 in the present manuscript. Here, the only difference is the timeframe used to describe the system. Waegeman et al

write $y(t-\tau)$, $y(t)$ and $x(t-\tau)$, Zhai et. al here write $y(t)$, $y(t + \tau)$ and $u(t)$, but both ideas seem fundamentally identical except for notation. Therefore, also point (2) is not novel in itself.

Finally, Waegeman et al. also already used "random" inputs, which corresponds to point (3), the use of stochastic control signals, in the present work.

Therefore, the present work might be accurately described as an extension of the work of Waegeman et al with partial observations.

This is not meant to reduce the quality of the presentation. The authors have clearly investigated the problem in a more thorough and robust manner than previous works, and they show many examples. In an ideal world, a thorough study should receive the credits it deserves, irrespective of any notion of novelty. Sadly, this is not yet the state of academic research.

I have several minor comments regarding some aspects of the manuscript, that could further improve the presentation:

- In the supplementary material, the parameter α is described as the "leakage rate" but also that "adjusts the learning rate". The latter point is confusing. While α is typically used as the learning rate parameter in many machine learning applications, it is not clear why the leakage rate would have the same function here.

- The input is written as $i(1)$, $i(2)$, ... but the reservoir equation itself is seemingly given on continuous time ($t + \Delta t$). First, it is strange to write a map-like equation (S0.1) but then imply that the time step has any importance. The dynamics should be independent of Δt except for stretching the timeline. It may be assumed that the authors want to say something along the lines of "input is held constant for $1/(\Delta t)$ steps", but the current notation is confusing. I advise to letting t strictly be an integer, and explain the input procedure if $i(1)$ etc. in that notation.

- Throughout the manuscript and supplementary, three different types of error measures are used: RMSE, "control success rate" and mean absolute error. Switching between these rates is making comparisons difficult. The MAE should be replaced by RMSE. And accuracy could be replaced by average RMSE or the root of the average MSE across the 16 target trajectories.

Reviewer #3 (Remarks to the Author):

I find this paper to be really very interesting and important. They adapt a fast trending concept of machine learning for forecasting dynamical systems, which is a random neural network architecture called reservoir computing, to problems of control. Control of Chaos was an important and highly cited topic in the 1990s through the early 2000s. It was based on various schemes of detailed local models. Recent trends in other directions have been data driven based on spectral operator theory. This current work has a simpler and perhaps more powerful method to all prior works, capable of solving similar problems, and perhaps more complex ones because of the approaches simplicity. I am really quite impressed how well it works. They demonstrate very nicely with a rigid rod problem capable of tracing arbitrary patterns and that alone has important implications in robotics. They further demonstrate the utility in several other interesting chaotic models. The paper is well written and I would say it should be published in its current form.

Summary of main changes

In response to the referees' comments, we have

1. provided extensive clarifications and explanations of our machine-learning controller,
2. added an additional example of tracking dynamical trajectories from a high-dimensional chaotic system,
3. unified the error measures and regenerated the affected figures, and
4. updated the simulation codes in GitHub.

Point-by-point response to referee comments

Report of Referee #1

General Comment: *“The aim of this study is to develop a model-free nonlinear control method for tracking periodic and chaotic motions of a two-arm robot. Unlike traditional control methods relying on prior knowledge and equations about the target system to be controlled, the authors construct a machine learning model to predict appropriate control inputs from observed states of a target system. They demonstrate that the proposed method works well for tracking complex trajectories generated by nonlinear and chaotic dynamical systems. Simulation results show that the method is tolerant against reasonably weak noise and disturbance.*

The basic idea behind the proposed model is found in the previous works [54,55] on tracking control, where an inverse model mapping a system state to a control input is approximated by a reservoir computing model. In this study, some advances are achieved. First, the controller works with data obtained by partial observations, by which the effort for measurements of system states can be reduced. Second, a pair of the present state and the one-step-ahead state is used as the input to the reservoir, which is new and reasonable for learning the association between them. Third, stochastic signals are used as control inputs in the training phase. The numerical results show that these factors are beneficial in successful tracking control.

The manuscript is well written, and the descriptions are informative. The methods and results are interesting overall, but there are some points to be addressed.”

Response: We appreciate the referee’s positive assessment of our work. The points raised by the referee have been fully addressed in the revised paper.

Comment 1: *“The authors call “ $y(t+dt)$ ” time-delayed observation. I don’t know why this is called “time-delayed” because a time-delayed state normally indicates “ $y(t-dt)$ ” in dynamical systems theory. In addition, it is questionable how $y(t+dt)$, which has not yet observed at time t , can be used for the update of the system at time t . It is better to explain how the observation process and training process are performed in practice, e.g. the order of the processes. If a storage of observed states is necessary, it might be a demerit of the proposed method compared to other online control methods. This point should be discussed.”*

Response: We agree with the referee that the term “time-delayed” used in the previous version can be confusing. Indeed, the quantity $y(t + dt)$ is the observed state vector at the next time step, if the current vector is $y(t)$. Both vectors are used only during the training phase where the observed time series are available, so the measurements of the observable state vectors in the immediate future are always available. In the testing or real-time control phase, the vector $y(t + dt)$ is no longer needed: it is replaced by $y_d(t)$, the desired state from the target trajectory to be tracked.

We have changed the terminology “time-delayed” to “immediate future” throughout the text. We emphasize that $y(t + dt)$ is needed only in the training phase. After training, the parameters of the machine-learning controller have been determined and are fixed, so our control scheme is not an online learning method. That is, in the actual operation or testing phase, at time t , $y(t + dt)$ is unobservable, but it is not needed because it is replaced by the desired $y_d(t)$ from the reference trajectory.

Furthermore, to avoid confusion, we have rewritten a large portion of the second paragraph in Discussion on page 8. The revised part reads

- Our inverse controller design method requires the observed state vector and its immediate future as input to the neural network in the training phase. The testing or deployment phase requires a combination of the current and desired output measurements: no future measurements are needed. More specifically, in the training phase, the input to the reservoir neural network consists of two vectors of equal dimension: (a) the observed vector from the robotic manipulator and (b) its immediate future version. This design enables the controller to naturally associate the second vector with the immediate future state of the first vector in the testing phase and to generate control signals based on this association. After training, the parameters of the machine-learning controller are fixed for testing, which distinguishes our control scheme from online learning. The controller in the testing phase is deployed to track a desired reference trajectory since the immediate future vectors $\mathbf{y}(t + dt)$ are replaced by the states generated from the desired reference trajectory, which are recognized by the machine as the desired immediate future states of the robotic manipulator to be controlled. The control signal generated in this manner compels the manipulator to imitate the dynamical system that generates the reference trajectory, resulting in precise tracking. We also take advantage of stochastic control signals for training the neural network to enable it to gain as much dynamical complexity as possible.

Comment 2: *“As shown in Fig. 1(c), the stochastic signals are used as control inputs. To learn appropriate associations between the present state and the one-step-ahead state, a large amount of training data (i.e. a long training length) seems to be necessary for the random-walk trajectory of the two-arm robot to fully cover the two-dimensional space. It is unclear how the authors determine the training length and how the training length affects the tracking performance. In addition, it is not obvious how much generalization ability of the model is acquired by the proposed method, i.e. whether the trained model has the ability to track unknown trajectory that was not used in the training phase.”*

Response: We have revised the section entitled “Effect of varying training parameters” in Supplementary Note 4 to explain how the hyperparameters affect the training performance. The four key hyperparameters are the training length T_{train} , network size N , reset time T_{ep} , and the amplitude η of the stochastic and uniformly distributed control signal employed in the training phase. In general, a larger network will lead to better tracking performance, while the training length has little effect on the performance. In addition, appropriate combinations of T_{ep} and η should be chosen to guarantee satisfactory performance. To illustrate these results as clearly as possible, we have improved three figures: Figs. S7 and S8 in this section as well as Fig. S11 in the section entitled “Robustness against variations” in Supplementary Note 5.

On the determination of the training length, our computations showed that, insofar as the training length is sufficiently long, which allows the system to explore a variety of states and dynamics, the performance of the machine-learning controller can be guaranteed. The following description in the section entitled “Effect of varying training parameters” in Supplementary Note 4 has been added:

- It can be seen that a larger network will lead to better tracking performance, while the training length T_{train} , insofar it is reasonable, has little effect on the performance. Likewise, the changes in the training reset time T_{ep} and different values of the amplitude η of the stochastic control input used in training, insofar as they are in a reasonable range, do not have a severe effect on the training performance. Figures S8(a-c) show the performance with respect to varying the reset time T and the amplitude η of the stochastic control input signal employed in the training phase for the circular, chaotic Mackey-Glass system with $\tau_{\text{mg}} = 17$ and the eight-figure reference trajectories, respectively. It can be seen that appropriate

combinations of T and η should be chosen to guarantee satisfactory performance. We emphasize that our goal is not to obtain an average well performance among the 50 trails of training. On the contrary, we aim at choosing a reservoir computer trained with optimal hyperparameters and applying it in the experiment. Similar to reinforcement learning, the well-trained reservoir computer is an intelligent agent and is able to handle the different tracking tasks in practice.

The generalizability of the control method can be argued, as follows. In the training phase, no specific trajectory is used. Rather, training is accomplished by using a stochastic control signal to generate a random-walk type of trajectory that “travels” through the entire state-space domain of interest. The machine-learning controller does not learn any specific trajectory example but a generic map from the observed state at the current time step to the next under a stochastic control signal. The training process determines the parameter values for the controller, which are fixed when it is deployed in the testing phase. The required input for testing is the current observed state $\mathbf{y}(t)$ and the desired state $\mathbf{y}_d(t)$ from the reference trajectory. The so-designed machine-learning controller is capable of making the system to follow a variety of complex periodic or chaotic trajectories to which the controller is not exposed during training.

We have added the following paragraph in the main text (on page 7) to explain the generalizability of our machine-learning control method:

- Our machine-learning controller, by design, is generalizable to arbitrarily complex trajectories. This can be seen, as follows. In the training phase, no specific trajectory is used. Rather, training is accomplished by using a stochastic control signal to generate a random-walk type of trajectory that “travels” through the entire state-space domain of interest. The machine-learning controller does not learn any specific trajectory example but a generic map from the observed state at the current time step to the next under a stochastic control signal. The training process determines the parameter values for the controller, which are fixed when it is deployed in the testing phase. The required input for testing is the current observed state $\mathbf{y}(t)$ and the desired state $\mathbf{y}_d(t)$ from the reference trajectory. The so-designed machine-learning controller is capable of making the system to follow a variety of complex periodic or chaotic trajectories to which the controller is not exposed during training. (Supplementary Notes 2 and 4 present many additional examples.)

Further support for the generalization ability of our control method can be found in Supplementary Note 5, in which the issues of safe region of initial conditions for control success, tracking speed tolerance, and robustness against variations in training parameters are addressed.

Comment 3: *“The authors demonstrate that the proposed method works well for 15 different kinds of target dynamical systems in the manuscript and the supplementary file. All these target systems are low-dimensional nonlinear dynamical systems and therefore can be regarded as toy problems, although the experiments are exhaustive. It is better to show that the proposed method is applicable to more practical case. It is also desirable to compare the proposed method with other tracking control methods in terms of tracking accuracy and computational efficiency.”*

Response: Thanks for the insightful comment. As explained, the training of our machine-learning controller is independent of any periodic or chaotic reference trajectories. The examples studied in fact contain examples of high-dimensional dynamical systems: the Mackey-Glass system for two values of the time delay. Because the system is described by a delay differential equation in which the state at time t depends on all the states in the

time interval $[t - \tau, t]$, the underlying phase-space dimension is infinite. In numerical computations, truncation is needed but the resulting system is still quite high-dimensional.

To further demonstrate the applicability of our control method to track trajectories from high-dimensional dynamical systems, we have studied an additional high-dimensional system: the 40-dimensional Lorenz-96 climate model. This system is a paradigm in atmospheric dynamics with rich complexity and serves as a more stringent test of the capabilities of the control method. The description of the model is presented in Supplementary Note 2, which reads

- The Lorenz-96 model [15] is a widely used benchmark dynamical system in meteorological and climatology research. It provides an abstract but effective representation of the atmospheric dynamics and exhibits complex behaviors similar to that observed in real atmospheric phenomena. The high-dimensional state vector is $X_{l96} = [x_{l96}^0, \dots, x_{l96}^{N-1}]^T$ with the governing equation [16]

$$\frac{dx_{l96}^n}{dt} = (x_{l96}^{n+1} - x_{l96}^{n-2})x_{l96}^{n-1} - x_{l96}^n + F \quad (1)$$

for $n \in \{0, 1, \dots, N - 1\}$, where the boundary conditions are $x_{l96}^{-1} = x_{l96}^{N-1}$, $x_{l96}^{-2} = x_{l96}^{N-2}$ and F is the forcing amplitude. We set $N = 40$ and $F = 8$. Since the two-arm manipulator is in a 2D plane, we choose the projection of the trajectory in the first two dimensions as the reference trajectory scaled into the respective ranges $[-0.6, 0.6]$ and $[-0.7, 0.7]$.

The results of tracking control of trajectories from the Lorenz-96 model are presented in Supplementary Note 4 with a new figure (Fig. S3), demonstrating that our machine-learning controller is capable of tracking trajectories from high-dimensional nonlinear dynamical systems. The following description has been added

- To further demonstrate the generalization ability of our controller, we study an additional high-dimensional nonlinear chaotic system: the Lorenz-96 system. Since our two-link robot arm manipulator operates in the 2D plane, we use the first two dynamical variables of the system to define a reference trajectory. Figure S3 shows the results of tracking this chaotic Lorenz-96 reference trajectory. Again, the machine-learning controller is trained with random-walk dynamics using a stochastic input control signal. In the testing phase, the controller is able to track the given complex trajectory without any fine-tuning or modification in the parameters.

In developing a machine-learning controller to track arbitrarily complex trajectories, we tested classical controllers such as PID (proportional-integral-derivative), MPC (model predictive control), and H_∞ trackers. In terms of computational complexity, these classical controllers are extremely efficient, while the training of our machine-learning controller with a stochastic signal can be quite demanding. However, there is a fundamental limitation with the classic controllers: such a controller can be effective only when its parameters were meticulously tuned for a specific reference trajectory. For a different trajectory, a completely different set of parameters is needed. That is, when the parameters of a classic controller are set for one trajectory, in general it cannot be used to track any alternative trajectory. In contrast, our machine-learning controller overcomes this limitation: it possesses the remarkable capability and flexibility to track any given trajectory after a single training session! This distinctive attribute sets our approach apart from conventional methods, so a direct comparison with these methods may not be meaningful. In fact, to our knowledge, the ability to adapt seamlessly to diverse trajectories makes our machine-learning controller unparalleled at the present.

We have added the following paragraph at the end of Discussion section to address the issue of comparison with existing methods:

- Finally, we remark that there are traditional methods for tracking control, such as PID, MPC (model predictive control), and $H\infty$ trackers (see [20,21], references therein). In terms of computational complexity, these classical controllers are extremely efficient, while the training of our machine-learning controller with stochastic signals can be quite demanding. However, there is a fundamental limitation with the classic controllers: such a controller can be effective only when its parameters were meticulously tuned for a specific reference trajectory. For a different trajectory, a completely different set of parameters is needed. That is, when the parameters of a classic controller are set, in general it cannot be used to track any alternative trajectory. In contrast, our machine-learning controller overcomes this limitation: it possesses the remarkable capability and flexibility to track any given trajectory after a single training session! This distinctive attribute sets our approach apart from conventional methods, so a direct comparison with these methods may not be meaningful. In fact, to our knowledge, the ability to adapt seamlessly to diverse trajectories makes our machine-learning controller unparalleled at the present.

Report of Referee #2

Major Comment: *““Model-free tracking control of complex dynamical trajectories with machine learning” studies the use of reservoir computing for control applications. A reservoir computer is used to predict a necessary control signal to steer a target dynamical system into a desired trajectory. Here, the authors use noise in training, and demonstrate the process in a large set of target dynamics.*

The main idea is sound and clearly works, despite some insufficient explanations in the supplementary materials. The result can be useful for many applications, but I worry about the novelty of the result.

The authors identify 3 major aspects:

- (1) use of only partial observations*
- (2) time-delayed input configuration for machine learning*
- (3) stochastic signals as the input control signals*

While the combination of these aspects is new, individually these have already been explored. It is known that reservoir computers can infer the state of unobserved variables, such as in the case of partial observations. For example, the work cited as Ref. 40 in the present manuscript presents such a case.

On the other hand, the basic training setup as shown in Fig 1 seems identical to that presented in Figure 2 of “Feedback Control by Online Learning an Inverse Model” (2012) by Waegeman, wyffels and Schrauwen, which is also cited as Reference 55 in the present manuscript. Here, the only difference is the timeframe used to describe the system. Waegeman et al write $y(t-\tau)$, $y(t)$ and $x(t-\tau)$, Zhai et. al here write $y(t)$, $y(t + \tau)$ and $u(t)$, but both ideas seem fundamentally identical except for notation. Therefore, also point (2) is not novel in itself. Finally, Waegeman et al. also already used “random” inputs, which corresponds to point (3), the use of stochastic control signals, in the present work. Therefore, the present work might be accurately described as an extension of the work of Waegeman et al with partial observations.

This is not meant to reduce the quality of the presentation. The authors have clearly investigated the problem in a more thorough and robust manner than previous works, and they show many examples. In an ideal world, a thorough study should receive the credits it deserves, irrespective of any notion of novelty. Sadly, this is not yet the state of academic research.”

Response: We thank the referee for this insightful and important comment. We agree with him/her that each of the three aspects stated in our previous version, individually, may not be regarded as new. We are grateful to the referee for his/her pointing out that “the combination of these aspects is new.”

As the referee pointed out, the 2012 work by Waegeman, wyffels and Schrauwen (cited as Ref. [55] in our previous version) is indeed the most relevant. The work treated the problem of stabilizing a dynamical system, such as a double pendulum, about a steady state using one control signal (a torque to the first pendulum, while no torque was applied to the second pendulum). The phase-space region to realize control is thus localized. This was effectively a regularization problem that aims to bring the system to a desired steady state and keep it there. Our tracking problem entails following a dynamic and time-varying (even chaotic) trajectory in the whole phase space. In our study, the controller needs not only to respond to disturbances and system variations but also to adjust the control inputs to make the system output follow a continuously changing reference signal.

The referee is completely correct that our input configuration for training is not new - in fact the same configuration appeared in Ref. [55] in the previous version. In addition, we wish to point out that it has also been previously known that a reservoir computer is able to learn the intrinsic dynamics of the system and infer the state of unobserved system variables, as demonstrated in Ref. [40] in the previous version. Our previous

statement about the novelty of our work was indeed not accurate - we are grateful to the referee for pointing this out.

We have done the following to fix/improve the presentation.

First, we have removed the previous inappropriate statement: ~~“Our approach goes beyond the previous works on reservoir computing based tracking control in three significant aspects: (1) use of only partial observations, (2) time-delayed input configuration for machine learning, and (3) stochastic signals as the input control signals.~~

Second, we have added the following description of the main contributions (in the penultimate paragraph in Introduction on page 2):

- Our control framework combines the following three features: (1) requirement of only partial state observations for both training and testing, (2) a training configuration for machine learning that involves the observed vectors at two consecutive time steps: $\mathbf{y}(t)$ and $\mathbf{y}(t + dt)$, and (3) use of a stochastic signal as the input control signal for training. With respect to feature (1), we note that reservoir computing was shown to be able to learn the intrinsic dynamics of the system and infer the state of unobserved system variables [40]. Moreover, features (2) and (3) were previously used in machine-learning stabilization of a dynamical system [55], where the phase-space region to realize control is localized. This was effectively a regularization problem that aims to bring the system to a desired steady state and keep it there. Our tracking problem entails following a dynamic and time-varying (even chaotic) trajectory in the whole phase space, where the controller needs not only to respond to disturbances and system variations but also to adjust the control inputs to make the system output follow a continuously changing reference signal. As we will demonstrate, our control scheme brings these features together to enable continuous tracking of arbitrary complex trajectories.

To further distinguish the two primary issues in control, we have added the following statement at the beginning of Discussion:

- The two main issues in control are: (1) regularization, which involves designing a controller so that the corresponding closed-loop system converges to a steady state, and (2) tracking - to make the output of the closed-loop system track a given reference trajectory continuously. In both cases, the goal is to achieve optimal performance despite disturbances and initial states [61].

We have also rewritten most of the second paragraph in Discussion to summarize the features of our machine-learning controller - please see our Response to Comment 1 of the first referee.

Minor Comment 1: *“I have several minor comments regarding some aspects of the manuscript, that could further improve the presentation.*

In the supplementary material, the parameter α is described as the “leakage rate” but also that “adjusts the learning rate”. The latter point is confusing. While α is typically used as the learning rate parameter in many machine learning applications, it is not clear why the leakage rate would have the same function here.”

Response: The referee is correct that learning rate is different from ‘leakage rate. While both parameters play a critical role in the speed and stability of learning, they serve different purposes and are used in different contexts. Specifically, the learning rate is for the optimization process to minimize the loss function, while the leakage rate is used to control the memory of the reservoir computing dynamical system. The following clarifications have been made in Supplementary Note 1.

First, below Eq. (S1), we wrote

- where α is the leakage parameter that determines the rate of “leakage” or “forgetting” in reservoir state updating,

Three lines down, we added

- It is worth mentioning that a smaller leakage rate α enables the system to retain more information about past inputs, whereas a larger one makes the system more biased towards recent inputs.

Comment 2: “The input is written as $i(1), i(2), \dots$ but the reservoir equation itself is seemingly given on continuous time ($t + \delta t$). First, it is strange to write a map-like equation (S0.1) but then imply that the time step has any importance. The dynamics should be independent of δt except for stretching the timeline. It may be assumed that the authors want to say something along the lines of “input is held constant for $1/(\delta t)$ steps”, but the current notation is confusing. I advise to letting t strictly be an integer, and explain the input procedure if $i(1)$ etc. in that notation.”

Response: The following clarification of Eq. (S1) has been added in Supplementary Note 1:

- Activated by the sequence of reservoir input signals $[\mathbf{i}(1), \mathbf{i}(2), \dots, \mathbf{i}(t)]$, the hidden layer state is updated step-by-step at the input signal time interval according to

$$\mathbf{r}(t + 1) = (1 - \alpha)\mathbf{r}(t) + \alpha \cdot \tanh [\mathcal{A} \cdot \mathbf{r}(t) + \mathcal{W}_{\text{in}} \cdot \mathbf{i}(t) + \mathcal{W}_{\text{bias}}], \quad (2)$$

Comment 3: “Throughout the manuscript and supplementary, three different types of error measures are used: RMSE, “control success rate” and mean absolute error. Switching between these rates is making comparisons difficult. The MAE should be replaced by RMSE. And accuracy could be replaced by average RMSE or the root of the average MSE across the 16 target trajectories.”

Response: To facilitate comparisons, we have changed the terms “control success rate” and “mean absolute error” into RMSE except in the Section entitled “Safe region of initial conditions for control success” in Supplementary Note 5. The reason is that, when the initial arm positions are randomly selected, the probability of successfully controlling a trajectory is typically far less than 100% (e.g., 50%). However, if the end effector is initialized in the third or fourth quadrant, the probability of successfully controlling a trajectory tends to increase dramatically and can often be close to 100%. Because of this, using only RMSE to characterize tracking control performance is inadequate.

Because of the change of “control success rate” to RMSE, three figures need to be regenerated: Figs. S7, S8, and S11. Overall, the results show that the ensemble-averaged RMSEs are relatively large, but the trend of RMSE changes in the parameter space can still be clearly identified. The reason for the relatively large RMSEs lies in training an ensemble of reservoir computers with the same set of hyperparameters and selecting the best-performed one. Insofar as we select the reservoir computer with the best training performance, the errors associated with other realizations matter little even when the average training error is large. The following explanation has been added into the section entitled “Effect of varying training parameters” in Supplementary Note 4:

- We emphasize that our goal is not to obtain an average well performance among the 50 trails of training. On the contrary, we aim at choosing a reservoir computer trained with optimal hyperparameters and

applying it in the experiment. Similar to reinforcement learning, the well-trained reservoir computer is an intelligent agent and is able to handle the different tracking tasks in practice.

About a possible change from the mean absolute error (MAE) to RMSE, we note that, since MAE measures the average of the absolute differences between the predicted and true values while the RMSE is the square root of the average of squared differences between the predicted and true values, it is not possible to directly change MAE to RMSE or vice versa. Our solution is training the neural networks again and using the RMSE as the measurement. As a result, Fig. S13 was revised, so was the relevant description in Supplementary Note 6.

Report of Referee #3

General Comment: *“I find this paper to be really very interesting and important. They adapt a fast trending concept of machine learning for forecasting dynamical systems, which is a random neural network architecture called reservoir computing, to problems of control. Control of Chaos was an important and highly cited topic in the 1990s through the early 2000s. It was based on various schemes of detailed local models. Recent trends in other directions have been data driven based on spectral operator theory. This current work has a simpler and perhaps more powerful method to all prior works, capable of solving similar problems, and perhaps more complex ones because of the approaches simplicity. I am really quite impressed how well it works. They demonstrate very nicely with a rigid rod problem capable of tracing arbitrary patterns and that alone has important implications in robotics. They further demonstrate the utility in several other interesting chaotic models. The paper is well written and I would say it should be published in its current form.”*

Response: We are grateful to the referee for recommending publication of our work.

REVIEWERS' COMMENTS

Reviewer #1 (Remarks to the Author):

All the comments have been appropriately addressed. It is ready for publication.

Reviewer #2 (Remarks to the Author):

The authors have replied to my previous criticism about the lack of novelty of their work, by refining their claims. I appreciate their efforts, however, regretfully they have not managed to sufficiently proof the novelty of their results when compared to the existing work of "Feedback Control by Online Learning an Inverse Model" by Waegeman, wyffels et Schrauwen, IEEE TNNLS 23, 10, 2012 in my view.

An essential part of their new argument is the differentiation between "static" and "dynamic" control, or "regularization" versus "tracking". However, the work by Wageman et al. already includes both cases as well. The first example in Waegeman et al.'s now 10-year-old paper is for a heated water tank, and the controller is clearly able to track a very complex trajectory (see Fig. 6 and 7 of that paper). This is a type of dynamic control. As pointed out by my previous reply, their method is essentially the same except for a difference in notation (the authors here use "future states", whereas Waegeman et al. used a shifted time-argument to make it easier to understand).

In addition, Waegeman et al. also used an online learning algorithm, which goes beyond what the authors of this submission are proposing.

The fact, that partial observations can be used in itself is not novel enough. As the reservoir contains memory, and by arguments along the line of Taken's embedding theorem, the reservoir can likely reconstruct a good approximation of the full phase space and thus operate as if it knows all the state variables. This is a known feature of reservoir computing. Once again, I appreciate the authors rigor and clear presentation, which I think is better than in the paper by Waegeman et al. However, in the substance, I do not believe that it sufficiently advances beyond the state of the art to justify publication in Nature Communications.

Irrespective of the above, the authors have addressed several of the critical technical comments of the previous round and the paper is now sound from a technical point.

Summary of changes: We are immensely grateful to Dr. Omelchenko for giving us the opportunity to revise the paper again to address the referee comments. Following Dr. Omelchenko's advice, we have extended the comparison to the literature to address remaining concerns of reviewer #2. We have also edited the manuscript to comply with the journal policies and formatting requirements.

Point-by-point response to the comments of reviewer #2

Comments: *“The authors have replied to my previous criticism about the lack of novelty of their work, by refining their claims. I appreciate their efforts, however, regretfully they have not managed to sufficiently proof the novelty of their results when compared to the existing work of ”Feedback Control by Online Learning an Inverse Model” by Waegeman, wyffels et Schrauwen, IEEE TNNLS 23, 10, 2012 in my view.*

An essential part of their new argument is the differentiation between ”static” and ”dynamic” control, or ”regularization” versus ”tracking”. However, the work by Wageman et al. already includes both cases as well. The first example in Waegeman et al.’s now 10-year-old paper is for a heated water tank, and the controller is clearly able to track a very complex trajectory (see Fig. 6 and 7 of that paper). This is a type of dynamic control. As pointed out by my previous reply, their method is essentially the same except for a difference in notation (the authors here use ”future states”, whereas Waegeman et al. used a shifted time-argument to make it easier to understand).

In addition, Waegeman et al. also used an online learning algorithm, which goes beyond what the authors of this submission are proposing.

The fact, that partial observations can be used in itself is not novel enough. As the reservoir contains memory, and by arguments along the line of Takens embedding theorem, the reservoir can likely reconstruct a good approximation of the full phase space and thus operate as if it knows all the state variables. This is a known feature of reservoir computing. Once again, I appreciate the authors rigor and clear presentation, which I think is better than in the paper by Waegeman et al. However, in the substance, I do not believe that it sufficiently advances beyond the state of the art to justify publication in Nature Communications.

Irrespective of the above, the authors have addressed several of the critical technical comments of the previous round and the paper is now sound from a technical point.”

Response: We thank the referee for his/her time to evaluate our revised manuscript and are grateful to him/her for stating that our paper “is now sound from a technical point.”

The referee was still concerned about the novelty of our work, especially when compared with the work by Waegeman, Wyffels, and Schrauwen. In that work, the authors proposed a feedback control scheme to solve a class of inverse problems with an online learning method using reservoir computing. Three relatively simple tasks were demonstrated to validate the scheme: controlling a nonlinear heating tank, linear flight pitch control, and balancing a double inverted pendulum. In our previous response, we stated that the tasks were of the regularization type in traditional control engineering, which can be readily accomplished in our framework of tracking control. In particular, the three systems were either linear or low-dimensional nonlinear systems, while our control framework is designed to control planar trajectories from nonlinear and chaotic systems in arbitrarily high dimensions, including those with an infinite-dimensional phase space.

A unique feature of our work that goes beyond the previous work is that our control framework requires only partial state observation. While, as the referee pointed out, the classical Takens' delay-coordinate embedding methodology can be used to construct the full phase space from partial observation, the reconstructed state is

equivalent to the original system but only in a topological sense: there is no exact state correspondence between the reconstructed and the original dynamical systems. For reservoir-computing based prediction and control tasks, such an exact correspondence is required, rendering inapplicable Takens embedding theorem. To our knowledge, achieving tracking control based on partial state observation is novel.

Another key aspect that distinguishes our work from Waegeman, Wyffels, and Schrauwen's lies in our unique training method. Different from their online learning algorithm, our learning approach is of the offline type. Online learning algorithms have problems such as instability, modeling complexity as required for nonlinear control, and computational efficiency. For example, it is difficult for online learning to capture the intricate nonlinear dynamics, causing instability during control. Trajectory divergence is also a common problem associated with online learning control, where sudden and extreme changes in the state can occur. In fact, as the dimension and complexity of the system to be controlled increase, online learning algorithms tend to fail. In contrast, offline learning is computationally extremely efficient and allows for more comprehensive and complex model training with the minimum risk of trajectory divergence through repeated training.

We have further clarified the main contribution of our work (in the last paragraph in the Introduction), which reads

- In this paper, we tackle the challenge of model-free and data-driven nonlinear tracking of various reference trajectories, including complex chaotic trajectories, with an emphasis on their potential applications in robotics. In particular, we examine the case of a two-arm robotic manipulator with the control objective of tracking any trajectories while using only partially observed states, denoted as vector $\mathbf{y}(t)$. Our control framework has the following three features: (1) requirement of only partial state observation for both training and testing, (2) a machine-learning training scheme that involves the observed vectors at two consecutive time steps: $\mathbf{y}(t)$ and $\mathbf{y}(t + dt)$, and (3) use of a stochastic signal as the input control signal for training. With respect to feature (1), it may be speculated that the classical Takens' delay-coordinate embedding methodology could be used to construct the full phase space from partial observation. However, in this case, the reconstructed state is equivalent to the original system but only in a topological sense: there is no exact state correspondence between the reconstructed and the original dynamical systems. For reservoir-computing based prediction and control tasks, such an exact correspondence is required. To our knowledge, achieving tracking control based on partial state observation is novel. In terms of features (2) and (3), we note a previous work [55] on machine-learning stabilization of linear and low-dimensional nonlinear dynamical systems, where the phase-space region to realize control is localized. This was effectively an online learning approach. In general, online learning algorithms have difficulties such as instability, modeling complexity as required for nonlinear control, and computational efficiency. For example, it is difficult for online learning to capture the intricate complex nonlinear dynamics, causing instability during control. Trajectory divergence is another common problem associated with online learning control, where sudden and extreme changes in the state can occur. In fact, as the dimension and complexity of the system to be controlled increase, online learning algorithms tend to fail. In contrast, offline learning is computationally extremely efficient and allows for more comprehensive and complex model training with minimum risk of trajectory divergence through repeated training. Our tracking framework entails following a dynamic and time-varying (even chaotic) trajectory in the whole phase space, where the offline controller can not only respond to disturbances and system variations but also adjust the control inputs to make the system output follow a continuously changing reference signal. As we will demonstrate, our control scheme brings these features together to enable continuous tracking of arbitrary complex trajectories.